# Measuring the Effectiveness of the ‘Batch Operations’ Energy Design Pattern to Mitigate the Carbon Footprint of Communication Peripherals on Mobile Devices

**DOI:** 10.3390/s24227246

**Published:** 2024-11-13

**Authors:** Roberto Vergallo, Alberto Cagnazzo, Emanuele Mele, Simone Casciaro

**Affiliations:** 1Department of Innovation Engineering, University of Salento, Via per Monteroni, 165, 73100 Lecce, Italy; simone.casciaro@unisalento.it; 2Faculty of Engineering, University of Salento, Via per Monteroni, 165, 73100 Lecce, Italy; alberto.cagnazzo@studenti.unisalento.it (A.C.); emanuele.mele@studenti.unisalento.it (E.M.)

**Keywords:** carbon-aware computing, green software, batch operations, GPS, 4G, green IoT

## Abstract

The Internet of Things (IoT) is set to play a significant role in the future development of smart cities, which are designed to be environmentally friendly. However, the proliferation of these devices, along with their frequent replacements and the energy required to power them, contributes to a significant environmental footprint. In this paper we provide scientific evidences on the advantages of using an energy design pattern named ‘Batch Operations’ (BO) to optimize energy consumption on mobile devices. Big ICT companies like Google already batch multiple API calls instead of putting the device into an active state many times. This is supposed to save tail energy consumption in communication peripherals. To confirm this, we set up an experiment where we compare energy consumption and carbon emission when BO is applied to two communication peripherals on Android mobile device: 4G and GPS. Results show that (1) BO can save up to 40% energy when sending HTTP requests, resulting in an equivalent reduction in $CO_{2}$ emissions. (2) no advantages for the GPS interface.

## 1. Introduction

Computing devices have become an essential part of modern life. Throughout the day, we interact with digital resources, both in our professional and personal domains. Smartphones, in particular, have become an extension of our bodies, offering services that have become so ingrained in our daily routines that they also affect our emotions [1,2]. Similarly, Internet of Things (IoT) devices are integrated into our daily lives. These devices, from smart thermostats to wearable health monitors, offer seamless connectivity and automation that have become essential to managing everything from home environments to personal health [3].

These devices not only have a great impact in our lives but also on the environment. The intersection of IoT and sustainability begins to be discussed in current literature. Existing research offers a compendium of best practices addressing sustainability challenges within IoT frameworks. Maqbool et al. (2023) [4] propose a comprehensive set of recommendations for achieving sustainability in buildings through IoT technologies, emphasizing the crucial role IoT plays in enabling real-time energy usage data collection for sustainable construction. Davison et al. (2022) [5] emphasize the pivotal role of IoT in aligning with the UN Sustainable Development Goals, emphasizing the responsibility lying with engineers, developers, policymakers, and entrepreneurs in steering the technology’s impact towards either promoting sustainable development or perpetuating detrimental linear business models and widening societal disparities. Nanjundan et al. (2023) [6] introduce a foundational analysis that contributes to a deeper comprehension of scientific advancements in IoT applications and the potential ecological ramifications accompanying their proliferation. Additionally, several other studies offer recommendations for meeting sustainability standards within IoT infrastructures. Hu et al. (2022) [7] delve into the impact of advanced manufacturing technologies and their influence on fostering eco-sustainability through the adoption of green industrial IoT (GIIoT), elucidating how manufacturing entities can achieve Green Innovation (GI) by embracing GIIoT technologies. Similarly, Paul et al. (2022) [8] explore the application of Green IoT (G-IoT) in meeting post-pandemic sustainability needs in the U.A.E., particularly in eco-friendly and sustainable smart agriculture. Their projections foresee G-IoT significantly transforming our information ecosystem by minimizing energy consumption. Suckling et al. [9] explore the overall environmental impact of smartphones, with a focus on user habits and how they affect $CO_{2}$ emissions.

The environmental impact of a smartphones and IoT devices does not come solely from the supply of resources, production processes and disposal. Intensive user usage of the device, which involves more frequent charging cycles, also plays a significant role. These charging cycles require electricity, the production of which, if based on non-renewable sources, further contributes to $CO_{2}$ emissions (Figure 1).

There is no way to charge a smartphone exclusively using renewable energy. Household electricity mainly comes from the national power grid, and the percentage of that energy generated from renewable sources is highly variable. The most accessible option we developers have is to create energy-efficient apps. The field of Green Software Engineering (GSE) is focused on defining guidelines and best practices for developing software that minimize the environmental impact of computer systems. It acts to counteract climate change, recognizing that computer systems and IoT technologies, will become increasingly essential in our society, and consequently, they will account for a growing portion of total $CO_{2}$ emissions. Belkhir et al. [10] shown that Greenhouse Gas Emissions (GHGE) from the Information and Communication Technology (ICT) sector could represent 14% of global GHGE by 2040.

The concept of development guidelines has long been known and is illustrated by Design Patterns—models dedicated to solving general problems that may arise in the process of designing any program. In the case study of Green Software, we have explored on Energy Design Patterns which were designed to optimizing energy consumption. Among them, Batch Operations (BO) is particularly effective in reducing energy use by grouping and scheduling similar tasks together to minimize resource consumption.

The BO pattern deals with the problem of Tail Energy Consumption, a phenomenon that occurs due to the repeated powering on and off of hardware resources. Tail Energy Consumption is a significant problem in systems where hardware components are frequently powering on and off, causing inefficiencies and increased power consumption. Each time a resource (e.g., processors, network interfaces, storage devices) is powered on, it consumes an amount of energy, even before the target activity is performed. Similarly, each time of a resource is powered off, it has an energy cost. When tasks are processed individually, these energy costs accumulate, leading to what is known as Tail Energy Consumption.

The BO model mitigates this problem by scheduling the execution of tasks that require the same resource until a certain threshold is reached. This threshold can be an amount of time or a maximum number of scheduled tasks. In this way, the energy overhead associated with repeated power on and off is minimized (Figure 2).

The aim of this study is to provide a comprehensive, practical and scientific evaluation of BO actual effectiveness. The primary objective is to empirically demonstrate its efficacy when applied to communication peripherals. This research extends its applicability to diverse domains, addressing not only mobile devices—with the experimentation conducted on the Android platform—but also exploring the potential portability of the Batch Operations pattern to various IoT devices. This research highlights the importance of integrating energy-efficient practices into application development, promoting a more sustainable and environmentally conscious approach across different technological domains.

To summarize, the main contributions of this work are:Empirical evidence and scientific validation of the effectiveness of the BO pattern in energy-efficient software design, addressing both power consumption and carbon emissions.Quantitative measurement of energy consumption for 4G and GPS communication interfaces.Quantitative comparison of energy consumption between an application utilizing the BO pattern and one that does not, highlighting the impact of the pattern on overall energy efficiency.

Our application code and the results of the statistical analysis are publicly available on GitHub [11]. The rest of the paper is structured as follows. Section 2 provides a detailed state of the art. In Section 3 we detail the materials and methods used in our experiment, with a specific focus on the tools and the methodology utilized to extract data. Section 4 reports on the results of the experiment, including the calculations of carbon emissions and statistical analysis. Finally, in Section 6 we discuss the significance of the results and sketches the future works.

## 2. Related Work

The issue of reducing energy consumption and the carbon footprint within the ICT sector has become a central focus in the scientific community. Energy optimizations can be carried out on multiple levels: cloud and high-performance computing (HPC), environmentally friendly digital services and smartphone apps.

Zhang et al. (2021) [12] evaluated the energy efficiency of servers running memory-intensive applications, focusing on the impact of memory architectures (NUMA and SMP) and the relationship between memory usage, power consumption, and performance. Lastovetsky et al. (2023) [13] presented an overview of challenges associated with energy consumption in HPC platform. Jayaprakash et al. (2021) [14] have conducted a systematic review on the different methodologies that can be applied to reduce energy consumption in the cloud—firefly algorithm, whale optimization algorithm (WOA), particle swarm optimization (PSO) and genetic algorithm (GA). Vergallo et al. (2024) [15] introduced a new strategy to reducing the carbon footprint associated with training Artificial Intelligence (AI) algorithm in the cloud. The approach dynamically shifts AI workloads to regions of the world where green energy is most accessible at any given time.

Moving on environmentally friendly digital services, Vergallo et al. (2024) [16] presented a new carbon-aware methodological framework called Digital Green (DG) to assess the environmental impact of digitization business processes. DG aims to quantify the environmental costs associated with digital transformation, ensuring that sustainable digitization results in lower resource consumption relative to the complexity of the process being digitalized. Mainetti et al. (2012) [17] proposed a secure NFC micro-payment system for Android mobile phones. The same first author in 2023 [18] proposed a new sustainable peer-to-peer offline payments systems based on One-Time Programs paradigm [19]. The solution allows offline payments without the presence of Trusted Third Party or Blockchain consuming energy to validate transactions.

To make applications more sustainable, one of the main challenges is the reduction of tail energy waste. The tail energy phenomenon characterized by energy consumption during periods of device inactivity, specifically after the completion of an active task, like data transfer. Balasubramanian et al. (2009) [20] demonstrated and quantified the energy wasted during 3G, GSM and Wi-Fi communications. In particular, in 3G networks, approximately 60% of energy consumption is wasted as tail energy, which occurs during high-power states after the completion of data transfers. In GSM networks, the tail time is shorter but more energy is consumed during data transfer. Wi-Fi has a similar overhead to 3G. Kaup et al. (2013) [21] shown how the expanded accessibility and higher data rates of 3G/4G cellular networks coupled with the rising prevalence of mobile applications, significantly impact the perceived Quality of Experience (QoE) for end-users. Liu et al. (2011) [22] have explored various techniques, including advanced power management strategies and the implementation of low-power states during periods of inactivity. Researchers of this study entail the introduction of TailTheft, a strategic framework leveraging a Virtual Tail mechanism and a Dual Queue Scheduling algorithm. This framework aims to reallocate Tail Time for energy-efficient data preloading and delayed transfer, thereby resulting in a noteworthy reduction in energy consumption within cellular communication systems.

In addition to the tail energy problem, the problem of energy consumption of applications and how to optimize the energy resources of devices is very much felt in the scientific community. Cong et al. (2020) [23] proposed a comprehensive survey of techniques designed to optimize energy consumption in mobile devices within the realm of Mobile Edge Computing (MEC). Kennedy et al. (2012) [24] proposed a comprehensive survey of techniques and solutions for optimizing energy consumption in wireless devices, particularly those focused on multimedia applications. The survey covers techniques at different layers, including hardware, software, and network, aiming to extend battery life without compromising user experience. Nawrocki and Sniezynski (2020) [25] introduced an innovative adaptive task-scheduling system employing Machine Learning and context information to optimize energy consumption in mobile devices. The purpose is to enable optimal task scheduling between the device and the cloud, considering contextual factors such as network connection type, location, execution time and cost, thereby enhancing efficiency in mobile systems. Bedregal et al. (2013) [26] provides an analysis of application-level energy consumption using the “Power Tutor” tool. This tool offers detailed information on the phone components with the highest energy consumption for each application. Li et al. (2014) [27] conducted a small-scale empirical evaluation of commonly suggested energy-saving and performance-enhancing coding practices. The study compares the energy efficiency of optimized versus unoptimized code, focusing on practices like network packet bundling, memory usage, array length access, direct field access, and invocation type. The findings show that certain practices significantly reduce energy consumption, while others have minimal impact on energy savings. Cai et al. (2015) [28] implemented an Android framework called “DelayDroid” to reducing energy tail consumption by managing 3G/4G resource access at runtime. The authors tested their framework using three DelayDroid-app obtaining a reduction of 3G/4G communications tail time energy waste about 36%. Corral et al. (2015) [29] introduced energy improvements at the kernel level by analyzing the operating system’s work. The authors demonstrated that custom kernels with energy optimizations can reduce the battery consumption by up to 33% for isolated tasks and improve device’s overall performance by up to 16%. The works of Li et al. (2014) [27], Cai et al. (2015) [28] and Corral et al. (2015) [29] are all based on BO pattern as report in Cruz et al. (2019) [30]. In addition, Cruz et al. (2019) [30] proposed a comprehensive catalog of patterns that developers can use to improve the energy efficiency of their applications. The authors analyzed 1783 mobile applications to identify specific implementation patterns that effectively address common design issues, which often lead to performance degradation and increased energy consumption.

### Motivations

The production of electricity is still very often characterized by high $CO_{2}$ emissions. Since there is no way in hands of software engineers to force energy production towards renewable sources, it follows that mitigating the environmental impact requires better energy management and carbon-aware development within our applications. Nevertheless, although the topic of energy consumption is widespread in the scientific community, there are still many gaps.

The quantitative analysis presented by Balasubramanian et al. [20], which focuses on devices like the Nokia N95 and HTC, and the work by Kaup et al. [21], primarily aim to improve Quality of Experience (QoE) but fall short of addressing the energy consumption of communication peripherals in a detailed manner. Similarly, while Liu et al. [22] discusses energy efficiency at the protocol level in network environments, it does not provide a concrete solution for managing peripheral energy consumption. The approach proposed by Nawrocki and Sniezynski [25], which leverages AI to make decisions and potentially offload tasks to the cloud, also has its limitations. AI models need to be trained for each device and customized to user habits, which, although beneficial in some respects, results in additional energy consumption during the training phase. Finally, Bedregal et al. [26] offers best practices for improving battery life but does not specifically tackle the optimization of energy use in communication peripherals, leaving room for improvement in this area.

The use of BO pattern is a possibility but there is currently a lack of scientific studies that qualitatively and quantitatively demonstrate the effectiveness of this technique. In addition, a quantitative analysis of the $CO_{2}$ consumption caused by smartphone use can lead developers to be more aware of the environmental impact of software from an energy point of view. The energy wasted during the frequent powering on and off of devices becomes a critical issue. This inefficiency not only shortens the battery life of individual devices, but also contributes to an increased environmental impact, compounding the global challenge of reducing carbon emissions.

It is therefore essential to bring to the attention of the scientific community, practitioners and software developers qualitative and quantitative values on the effectiveness of the BO model in terms of energy optimization and reduction of the resulting $CO_{2}$ emissions.

## 3. Materials and Methods

The method is to empirically assess energy consumption and resulting carbon emissions when mobile devices access radio and GPS resources. In Figure 3 we report the research methodology adopted in our work, which comprises 4 steps.

The first step includes the development of Android application capable of performing tasks both with and without the BO pattern. The second step includes an in-depth analysis of the available energy monitoring tools and design of a testing environment. The third step includes the definition of a mathematical carbon emissions model to quantify the application’s carbon footprint. The fourth step includes a statistical analysis to evaluate the collected results and determine the effectiveness of the BO pattern.

The experiment consists in connecting an energy monitoring tool to a device to get the energy consumption information of 4G and GPS interfaces, with and without BO pattern.

The experiment is divided into three phases. In the first phase, we will connect the device to the energy monitoring tool to get the initial battery status information. In the second phase, a mobile application will have to perform one of the following tasks:4G resource access with BO;4G resource access without BO;GPS resource access with BO;GPS resource access without BO.

During the execution of the chosen task, mobile application makes the call to the resource 24 times every 15 s. This execution is repeated a total of 1000 times to simulate a one-hour real-world intensive usage scenario. In the third phase, we will reconnect the device to the energy monitoring tool to get the battery status information at the end of the task performed in the previous phase. By comparing battery information before and after the execution of a task, we will be able to analyze consumption for each approach and resource. The experiment starts over from phase one until all the tasks in phase two have been done at least once. Once we have obtained the energy consumption of each task presented in phase two of the experiment, we will be able to conduct a statistical analysis of the consumption quantify the $CO_{2}$ emissions for each approach and resource.

### 3.1. Bo Pattern Implementation

As shown in Figure 4, the logical architecture of the application implementing the energy pattern includes a dedicated package responsible for receiving and executing operations based on a predefined energy policy. This architecture also includes simplified models of the two key resources under investigation–HTTP and GPS–by providing mock executions for these operations, which the energy pattern schedules. The management and scheduling of access to these key resources will be handled by the BO pattern.

In order to test the HTTP resource, we used OkHttp3 (https://square.github.io/okhttp/, accessed on 12 November 2024) library. As report in official documentation, OkHttp3 is a client-side library designed to be “Efficient by Default”. This means it not only simplifies implementation for developers but also automatically handles common connection-related issues, ensuring efficient and reliable HTTP communication with minimal additional configuration. In order to test the GPS resource, we used a native library into Google’s Mobile Services (GMS). GMS are a collection of APIs typically pre-installed on Android devices, which provides a set of services essential for basic operations.

To implement our application, we utilize structures from the Command design pattern. Focusing on two main classes: BatchOperationsManager, which acts as the invoker, and BatchOperation interface, which must be implemented by all operations using the service. In BatchOperationsManager, after importing the necessary HTTP and GPS libraries, we define the instance object and the static method getInstance(). The class includes two ArrayLists of BatchOperation-one for HTTP operations and the other for GPS operations-along with private attributes OkHttp3 client and fusedLocationProviderClient for GPS. A modified Singleton getInstance() method is used to pass an attribute to the constructor. A developer who wants to use the Energy Pattern “Batch Operation” needs to import the Package and perform a refactoring on the operations they want to execute. They must choose which methods they want to schedule and implement them in classes that implement the “Operation” interface, referring to the methods of the package for the execution of those operations (Figure 5).

A graphical user interface (GUI) was developed to facilitate the research process, allowing us to easily initiate the data collection process for energy consumption with and without the BO pattern. This interface facilitates the testing procedure by providing a user-friendly platform to control and monitor the execution of experiments.

### 3.2. Testing Environment

To accurately measure the effectiveness of the BO pattern, we utilized PowDroid [31], an energy monitoring tool designed to collect and manage energy consumption data. PowDroid works by capturing “Raw Battery Data” through various software components such as:*Batterystats*: a tool included in the Android framework that utilizes the Android Debug Bridge (ADB) to collect raw data on battery usage.*Bugreport*: another tool within the Android framework that generates a report file in zip format based on the results obtained from Batterystats.*Google Battery Historian*: an open-source tool used to convert the aforementioned zip file into a user-friendly CSV file. This tool makes it easier to analyze and interpret the battery usage data for improved understanding and user-friendliness.

The idea is to use PowDroid to collect raw energy consumption data from an application utilizing the BO pattern and compare it with the same application running without the pattern (Figure 6). By capturing energy usage metrics in both scenarios, we aim to conduct a thorough analysis to determine the effectiveness of the BO pattern in reducing energy consumption.

The resulting measurement file contains a table where each row represents a “battery-stat” associated with a series of data, including the start and end timestamps of the section, battery voltage, remaining charge, and other information. The Table 1 provides a more detailed overview of the information contained in the output file.

The device used for experimentation is a Samsung Galaxy A20e (SM-A202F/DS) with:Android version 10;Baseband version A202FXXU3BUB1;Display brightness set to 50%;Audio volume set to 50%;Google Play Services (only for HTTP tests) and the power-saving mode disabled.

The methods for the test cases were structured consistently across all cases to ensure high experimental accuracy and replicability. Each method creates an Operation object, representing a single call to the resource, and initiates execution 24 times, with each execution separated by 15 s. This process is enclosed within a loop of 1000 repetitions. The choice to oversize the number of executions ensures that requests continue to be sent without interruptions for the chosen duration (one hour in this specific case).

For the app with BO, an additional scheduling parameter is required. App with BO pattern adds each call to a list of scheduled tasks. Every 120 s the application will execute the tasks in the list. The list will always contain about 8 calls to the resource of interest. The choice of a 120 s is a good compromise between maintaining acceptable latency and saving energy. Shorter wait times could reduce the effectiveness of the BO, while longer wait times could significantly worsen the user experience.

To ensure consistent and controlled conditions, we established a baseline setup prior to running each experiment. The tests are conducted at specific battery levels (100%) to standardize initial conditions, and each run lasting one hour. Additionally, the device was positioned in an environment with stable signal strength to avoid variations in power consumption due to fluctuating network or GPS signal quality.

The following tests were performed on each resource (HTTP and GPS):Without Batch Operations: tasks are performed normally without some optimization strategy. This output represents the baseline used for comparison.With Batch Operations: tasks are performed using the BO pattern.

The PowDroid script is executed on a Windows device, connecting the mobile phone via USB Debug mode and granting necessary permissions. After collecting initial battery status information, the phone is disconnected, and the application is launched. For HTTP tests, the screen can be turned off to continue execution in the background, while GPS tests require the activity to remain active, so the impact of the display on power consumption must be considered.

At the end of the hour, the smartphone is reconnected to the PC, and the end of the execution is confirmed. PowDroid then retrieves the battery status information, processes it, and stores it in a CSV file for analysis. The CSV file follows the structure presented in Table 1.

### 3.3. CO_2_ Emissions Model

In addition to measuring energy consumption, it is important to assess the corresponding $CO_{2}$ emissions with and without the BO pattern. By comparing the $CO_{2}$ emissions data for both scenarios, we can determine if the implementation of the BO pattern leads to a significant environmental benefit, further validating its effectiveness in promoting sustainable application development.

To achieve this, we utilized Electricity Maps (https://app.electricitymaps.com/map, accessed on 12 November 2024), a service that provides real-time data on carbon consumption worldwide. This website offers both web access and an API for data retrieval and processing. In our case, we focused on obtaining the Carbon Intensity (CI) value, which indicates how many grams of $CO_{2}$ are released into the air to produce one kilowatt-hour (kWh) of electricity. The CI value depending on the time and geographical area. This data from Electricity Maps will be essential for evaluating the environmental impact and carbon footprint reduction achieved through the implementation of energy-efficient patterns like BO.

All tests were performed in our laboratory in the south of Italy during the month of April. However, we used historical data from the Electricity Maps website for the geographical area “Meridione (Italy)” during the month of April 2024 to obtain an “aggregated” Carbon Intensity (CI) value. This allows us to calculate the amount of $CO_{2}$ emissions (expressed in grams) associated with the test runs. The calculation is performed using the equation:(1)$$CO_{2}=EnergyConsumed\;\left[\mathrm{kWh}\right]\cdot CI\;\left[g/\mathrm{kWh}\right]$$
where “CI” refers to the Carbon Intensity value obtained from the website, and “$CO_{2}$” represents the grams of carbon dioxide produced.

## 4. Results

This section presents the detailed results from our experimental investigation, focusing on two critical aspects: energy consumption and $CO_{2}$ emissions into the atmosphere. This thorough analysis aims to reveal the tangible effects of integrating the Batch Operations energy design pattern into communication peripherals. We systematically examine energy consumption patterns to understand their impact on device efficiency. Simultaneously, we assess the environmental implications by evaluating the resulting carbon dioxide emissions. The findings offer comprehensive insights into both the practical performance and environmental impact of the Batch Operations pattern.

During the testing phase, the status of the 4G and GPS signals is report in Table 2 and Table 3.

### 4.1. Energy Consumption

After completing the tests, the next step is to interpret the results generated by the script. The descriptive metrics table of PowDroid (Table 1) is particularly helpful for this analysis. We focus on the energy consumed, measured in Joules, in each section of the testing process. By identifying the rows corresponding to our test run using the “Top App” information, we can determine the start and end timestamps, along with all relevant details. To calculate total energy consumption, we sum the values in the Energy column and compare the two scenarios—with and without the BO pattern. This comparison allows us to evaluate the effectiveness of the energy pattern in terms of reducing energy consumption. Figure 7 and Figure 8 displays the results of the cellular network and GPS tests, respectively, highlighting the impact of the BO pattern.

The total energy consumption of cellular network is 392.86 J with BO pattern and 643.59 J without BO pattern. For GPS, the total energy consumption is 2357.89 J with BO pattern and 2266.76 J without BO pattern.

In terms of energy consumption, GPS interface has higher demands in both scenarios. The GPS technology relies on signals from satellites in orbit which are in constant motion to the Earth. This implies that GPS must be activated more often and for longer sessions than an HTTP communication, leading to higher energy consumption. In addition, GPS requires continuous tracking to ensure accuracy, which significantly reduces the potential benefits of the BO approach.

### 4.2. $CO_{2}$ Emitted in the Atmosphere

Using historical data from Electricity Maps, the PowDroid output file, and Equation (Equation 1), we can calculate the amount of $CO_{2}$ emitted during our tests for both the application with and without the BO pattern. This calculation allows us to quantify the environmental impact of implementing the BO pattern in terms of carbon emissions, providing a clearer understanding of its effectiveness in reducing not just energy consumption but also its carbon footprint.

HTTP
(2)$$\begin{array}{cc} CO_{2}\left(NoBatch\right) & =EnergyConsumed\;[\mathrm{kWh}]\cdot CI\;[g/\mathrm{kWh}]\\ & =\frac{EnergyConsumed\;[J]}{3.6\cdot 10^{6}}\cdot CI\;\left[g/\mathrm{kWh}\right]\\ & =\frac{643.59}{3.6\cdot 10^{6}}\cdot 289\\ & =1.79\cdot 10^{-4}\cdot 289=0.0517\;g\\ CO_{2}\left(Batch\right) & =EnergyConsumed\;[\mathrm{kWh}]\cdot CI\;[g/\mathrm{kWh}]\\ & =\frac{EnergyConsumed\;[J]}{3.6\cdot 10^{6}}\cdot CI\;\left[g/\mathrm{kWh}\right]\\ & =\frac{392.86}{3.6\cdot 10^{6}}\cdot 289\\ & =1.09\cdot 10^{-4}\cdot 289=0.0316\;g \end{array}$$GPS
(3)$$\begin{array}{cc} CO_{2}\left(NoBatch\right) & =EnergyConsumed\;[\mathrm{kWh}]\cdot CI\;[g/\mathrm{kWh}]\\ & =\frac{EnergyConsumed\;[J]}{3.6\cdot 10^{6}}\cdot CI\;\left[g/\mathrm{kWh}\right]\\ & =\frac{2266.76}{3.6\cdot 10^{6}}\cdot 289\\ & =6.29\cdot 10^{-4}\cdot 289=0.181\;g\\ CO_{2}\left(Batch\right) & =EnergyConsumed\;[\mathrm{kWh}]\cdot CI\;[g/\mathrm{kWh}]\\ & =\frac{EnergyConsumed\;[J]}{3.6\cdot 10^{6}}\cdot CI\;\left[g/\mathrm{kWh}\right]\\ & =\frac{2357.89}{3.6\cdot 10^{6}}\cdot 289\\ & =6.54\cdot 10^{-4}\cdot 289=0.189\;g \end{array}$$

For HTTP, the application emitted 0.0517 g of $CO_{2}$ without the BO pattern, compared to 0.0316 g with it. In the GPS test, however, the results differed: without the BO pattern, the emissions were 0.181 g of $CO_{2}$, while with the pattern, they slightly increased to 0.189 g. Although BO significantly reduced emissions in the HTTP test, the GPS results showed no substantial improvement, and energy consumption remained comparable in both cases.

The percentage decrease and increase can be calculated using the Equations (Equation 4) and (Equation 5):(4)$$CO_{2}PercentageDecrease=\frac{CO_{2}\left(NoBatch\right)-CO_{2}\left(Batch\right)}{CO_{2}\left(NoBatch\right)}\times 100$$
(5)$$CO_{2}PercentageIncrease=\frac{CO_{2}\left(Batch\right)-CO_{2}\left(NoBatch\right)}{CO_{2}\left(Batch\right)}\times 100$$

In particular, a 39% reduction in $CO_{2}$ emissions was measured for HTTP (Equation (Equation 6)); a 4% increase in $CO_{2}$ emissions was measured for GPS (Equation (Equation 7)).

HTTP
(6)$$\begin{array}{cc} CO_{2}PercentageDecrease & =\frac{CO_{2}\left(NoBatch\right)-CO_{2}\left(Batch\right)}{CO_{2}\left(NoBatch\right)}\times 100\\ \; & =\frac{0.0517-0.0316}{0.0517}\cdot 100=39\% \end{array}$$GPS
(7)$$\begin{array}{cc} CO_{2}PercentageIncrease & =\frac{CO_{2}\left(Batch\right)-CO_{2}\left(NoBatch\right)}{CO_{2}\left(Batch\right)}\times 100\\ \; & =\frac{0.189-0.181}{0.189}\cdot 100=4\% \end{array}$$

Although the $CO_{2}$ savings per individual task may seem modest, when spread over a year of average daily use, it can result in a substantial reduction in emissions. For instance, when using the BO pattern for HTTP operations, the application saves 0.0201 g of $CO_{2}$ per hour. Projecting these data using Digital 2024 data by Data Reportal (https://datareportal.com/reports/digital-2024-global-overview-report, accessed on 12 November 2024) for an average usage of 6 h per day, this results in a $CO_{2}$ reduction of approximately 44 g per device per year. Extending this value to all smartphones in circulation (about 7 billion), this means an annual reduction of about 300,000 tons of $CO_{2}$ per year.

### 4.3. Statistical Analysis

In this section we want to prove that the differences found in experiments have statistical significance. We have used the data shown in the previous graphs for HTTP (Figure 7) and GPS (Figure 8). Analyzing the distribution of data we can see that the distribution is not Gaussian, for both HTTP Figure 9 and GPS Figure 10. We can conclude that we can’t use a *t*-Test for hypothesis test, because the assumptions are not verified for it.

In our study, we chosen a non-parametric hypothesis test, specifically the Mann Whitney U-Test [32], to determine if there is a statistically significant difference between the two methods: with and without the Batch Operation. We compared the data from both scenarios to see if they produce different results.

The U-Test was implemented using the mannwhitneyu function from the scipy library, with a *p*-value threshold set at 0.01 for stronger statistical significance. The implementation of the U-Test is public available on GitHub [11]. If the test returns a *p*-value less than 0.01, we reject the null hypothesis, indicating a significant difference between the methods. Otherwise, if the *p*-value is greater than 0.01, we accept the null hypothesis, suggesting no statistically significant difference between the two approaches. Table 4 show our results.

In both cases, the *p*-value obtained confirms a statistically significant difference between the use and non-use of the batch method. This indicates that the BO pattern has a measurable impact on the performance metrics evaluated, reinforcing its effectiveness in the scenarios tested. The statistical significance at the 0.01 level provides strong evidence that the observed differences are due to the influence of the batch processing approach.

## 5. Discussions

This study aimed to provide a comprehensive, practical, and scientific evaluation of the effectiveness of the Batch Operations (BO) pattern, particularly when applied to communication peripherals. Our primary objective was to empirically demonstrate its impact on energy efficiency. This research underscores the need for sustainable and environmentally conscious approaches across diverse technological domains, demonstrating the broader applicability and benefits of the BO pattern. Moreover, the obtained results underscore the critical role of software design in shaping the environmental footprint of electronic devices.

While the BO model is a relatively new concept and lacks scientific evidence supporting its efficiency, the paper contributes to bridging this gap by providing an empirical analysis of BO’s effectiveness in energy optimization, particularly for communication interfaces such as 4G and GPS.

In particular, the paper provides scientific evidence on the efficacy of the BO energy design pattern in optimizing energy consumption on mobile devices. Summing up the results obtained from the experimentation, we can see that the use of Batch Operation has enhanced performance associated with HTTP requests, whereas location operations have not benefited from it. In fact, findings indicate that implementing BO can yield significant energy savings of up to 40% when sending HTTP requests, although it appears to offer limited advantages for the GPS interface on Android devices. The limited advantages for the GPS interface could be caused to several intrinsic characteristics of GPS technology. First, the GPS technology relies on signals from satellites in orbit which are in constant motion to the Earth. To maintain accurate positioning, GPS requires continuous tracking to ensure precision. Second, the number and availability of GPS satellites can vary significantly based on time of day. This variability introduces additional challenges to implementing energy-saving techniques like BO. Third, the energy efficiency of GPS is highly dependent on GPS technology embedded within the device. Modern smartphones may utilize advanced GPS chips designed to optimize power consumption through techniques like adaptive tracking or sensor fusion.

In response to these limitations, several complementary methods for optimizing GPS energy consumption could be adopted. Ezzini & Berrada (2021) [33] propose a smart adaptive sampling method for GPS sensors using the accelerometer. This approach can reduce GPS sensing activity significantly during periods of low mobility, achieving considerable energy savings without major accuracy trade-offs. Chen et al. (2017) [34] propose a selective satellite tracking algorithm that provides similar positioning accuracy with a subset of selected satellites. The authors obtained an energy saving of 22.7%. Oliveira & de Mello (2018) [35] proposed an approach that schedules GPS and WiFi operations at regular intervals. This method reduces battery drain by up to 20% without compromising functionality.

To validate the results obtained, we employed a hypothesis test, specifically the U-Test, which confirmed the significant differences between the usage and not-usage of BO pattern. This analysis highlighted the substantial impact of using the BO pattern with both the HTTP protocol and GPS, demonstrating a clear statistical distinction in both cases. The U-Test provided strong evidence that the BO pattern leads to meaningful performance improvements, reinforcing the validity of our findings.

The study’s methodology incorporated detailed repetition testing, maintaining consistency across various test cases for accurate and precise analysis. Each method executed 24 times, with 15-s intervals, encapsulated within a loop of 1000 repetitions, effectively spanning an hour of continuous operation. This oversizing of executions ensured uninterrupted testing under predefined conditions, facilitating in-depth evaluation. To monitor energy consumption, we adopted specialized measurement tools for Android systems, the PowDroid script, to collect energy data. Additionally, to assess the impact on $CO_{2}$ emissions, the study employed Electricity Maps, which provides real-time global data on carbon intensity. Based on the measurements conducted, we expect that the overall performance achievable in real-world and daily usage contexts does not exceed the upper bound defined by the presented test results. Like the implemented model, the Design Pattern approach can be employed to implement BO from scratch within one’s application. It is essential to note that the pattern is already widely used by major IT companies, such as Google.

However, the BO pattern is not suitable for massive implementation. The implementation of BO is particularly effective in scenarios where operations can be grouped without compromising user experience. For instance, payment with mobile banking app, photos backup and data collection apps are suitable for the BO pattern because the operations to be performed do not require a real-time connection and are tolerant to delays. Instead, applications such as GPS navigation, messaging applications and social networks are not suitable for implementing the BO pattern because scheduling operations would result in a poor user experience. In addition, there are two limitations. The first, not all SDK and operating systems natively support the implementation of this pattern. This means that developers must manage and develop the scheduling of operations themselves, which can lead to major delays in the development of apps and updates. The second, it should be the operating system that natively manages resource scheduling but, as we said before, not all applications are suitable for implementing the BO pattern without compromising the user experience. For a more widespread adoption of BO, SDK and operating systems need to be improved, providing more robust and accessible tools for implementing this pattern. In addition, increased developer awareness of the benefits of BO and appropriate use cases would make the integration of BO into applications a standard practice.

This study focused on 4G and GPS peripherals, which are the most energy intensive for smartphones and IoT devices. The BO pattern can also be applied to interfaces such as Bluetooth and WiFi. Bluetooth connections are often used to transmit and synchronize data between smartphones and IoT devices asynchronously and for multimedia applications. For each connection established, data packets are forwarded to set the parameters of the connection—such as advertising, connection request and service discovery packets. However, Bluetooth, and in particular Bluetooth Low Energy (BLE), is already less energy intensive than 4G and GPS. This means that applying the BO pattern to asynchronous applications may result in only minimal energy savings, potentially negating efforts made to optimize the application. However, future research could explore how the BO pattern could be implemented to reduce electromagnetic interference caused by configuration packets exchanged between devices. Grouping and scheduling transmissions using the BO pattern would reduce the number of configuration packets exchanged between devices.

WiFi connections, on the other hand, are very common for periodic transmissions and data synchronization, such as background updates or backups. In this context, the BO pattern can group and schedule these transmissions, and reducing the frequency of radio channel activation and associated power consumption. Based on the data obtained in this paper, we expect the WiFi interface to perform similarly, if not better, than 4G. Both 4G and WiFi resources consume more energy each time they are activated. Therefore, reducing the activation frequency of the interface through BO pattern scheduling could lead to energy-saving behavior that justifies the optimization process.

### Threats to Validity

The main threat to internal validity is the controlled nature of the experimental scenario, which intentionally differs from real-world conditions to establish an upper bound of performance. The scenario is intentionally not representative of reality since the ultimate goal is to conduct an empirical analysis of BO and not propose a practical implementation for the model. It is evident that in normal device usage, numerous other factors need to be considered to evaluate the model’s effectiveness. To mitigate this risk, we performed a statistical analysis to verify that we had collected enough data. The analysis confirms the effectiveness of the proposed model in reducing energy consumption. The analysis provided mathematical validation, demonstrating that the implementation of the model leads to significant energy savings.

Another threat to internal validity concerns the GPS operations and the applicability of the BO pattern to this specific resource. The lack of expected results suggests potential issues. First, the intensive use of the GPS hardware may lead to device overheating, reducing energy efficiency. Second, the test environment, which required the screen to remain active, likely impacted energy performance negatively. These factors combined may have negated any energy savings from implementing BO, resulting in no overall improvement. To address this, we repeated GPS access a thousand times to obtain more reliable data, though GPS energy consumption remains a topic for future research.

The main threat to external validity is the usage of a single device. This decision was driven by the need for highly controlled conditions to isolate and clearly demonstrate the effects of the BO pattern on energy consumption. This focused approach allowed us to conduct a thorough and detailed analysis, setting the stage for future research. To mitigate this issue, we chose a device that was part of Samsung’s budget series, which saw strong overall sales due to its affordable pricing and reasonable specifications. Moreover, we suggest future research to include testing across various hardware platforms and operating systems, Android and iOS, to confirm the BO’s effectiveness in diverse environments. Finally, we suggest conducting these tests in different geographical locations and to study possible correlations between signal coverage and energy consumption. To make this work replicable, we have publicly shared our codes and data on GitHub [11].

## 6. Conclusions

Our work contributes to the field of energy optimization in IoT application development. Compared to baseline techniques, which often lack detailed consideration of energy consumption patterns specific to communication interfaces, our approach focuses specifically on energy patterns, with a particular emphasis on 4G and GPS communication interfaces. Differently from existing works, our research relies on empirical evidence and scientific validation to demonstrate the effectiveness of BO in optimizing software design for energy efficiency. This provides a solid and scientifically validated basis for performance evaluation. We also provide a focus on Communication Interfaces: while many techniques concentrate on overall software optimization, our model specifically targets communication interfaces like 4G and GPS, recognizing their significant impact on resource consumption and improving energy efficiency in realistic scenarios.

This work is useful for the following categories of stakeholders:Software and developers and practitioners are motivated to rethink the way they stack API calls, they also have a tool for Android platform that can be reused to batch any API call;IT companies can demonstrate commitment in the ecological transition by applying a concrete strategy in their project;Researchers have new challenges to face (see next paragraph);OS builders can incorporate batch processing within their products to facilitate the work of app developers;Chipset and device manufacturer can consider the results of this research to improve the performance of their products in case of heavy use.

Future developments are poised to expand the application of this methodology to verify the benefits of BO not only on smartphones from other vendors, such as Apple, but also, on a wider spectrum of IoT devices. Indeed, we encounter a generalization limitation that, as with many new proposals, bring limitations in generalizing the findings to all scenarios. Further research and testing may be required to explore the full scope of applicability, as it is very difficult, stemming from the results of our current research, to foresee if the BO energy pattern would behave similarly on different platforms and technologies. So future research will cover this knowledge gap. Moreover, the same type of testing can be extended to other peripherals, such as Wi-Fi, Bluetooth and various sensor interfaces, broadening the scope of investigation beyond the 4G and GPS interfaces. Extending this research to encompass devices from various manufacturers and IoT technologies will offer a more comprehensive understanding of the efficiency and adaptability of energy design patterns across diverse platforms. This expansion could significantly contribute to the broader implementation of energy-efficient strategies in diverse technological landscapes, enhancing sustainability and reducing the environmental impact across the spectrum of electronic devices and IoT ecosystems.

## Figures and Tables

**Figure 1 sensors-24-07246-f001:**
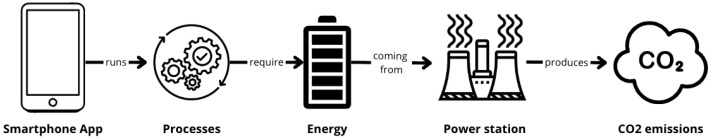
How smartphone apps contribute to $CO_{2}$ emissions through energy consumption.

**Figure 2 sensors-24-07246-f002:**
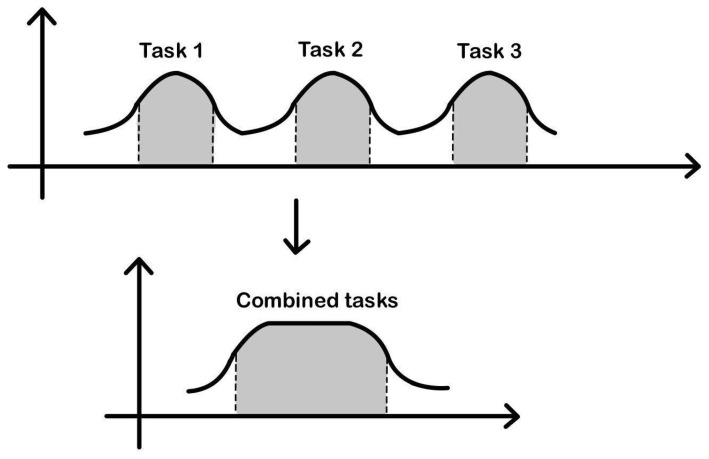
Tail Consumption graphical model.

**Figure 3 sensors-24-07246-f003:**
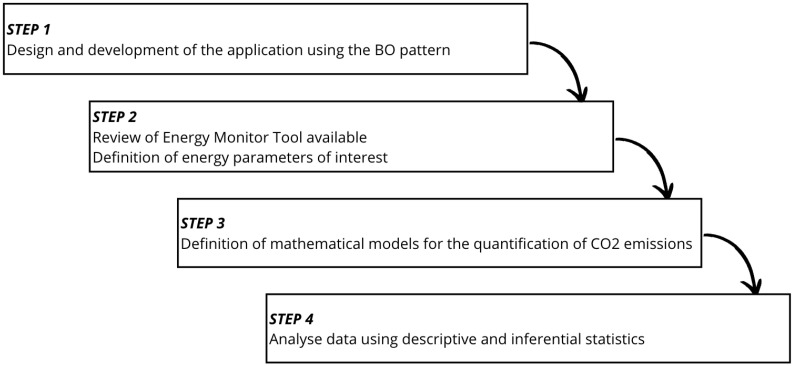
The research methodology adopted in this paper.

**Figure 4 sensors-24-07246-f004:**
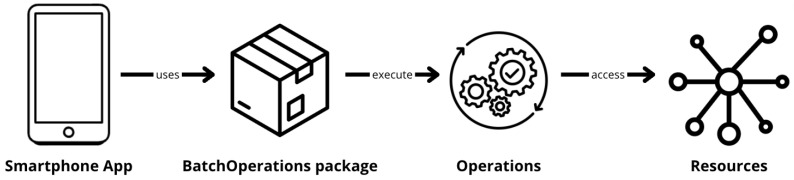
System logical architectures.

**Figure 5 sensors-24-07246-f005:**
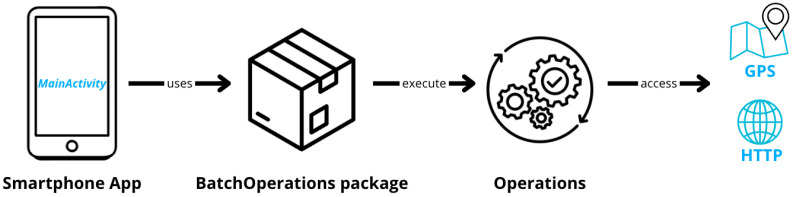
Physical architecture of the package.

**Figure 6 sensors-24-07246-f006:**
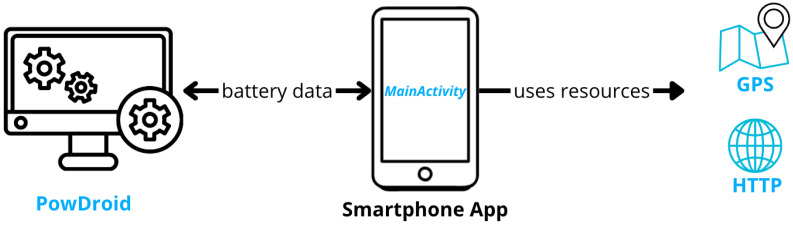
Physical architecture of test.

**Figure 7 sensors-24-07246-f007:**
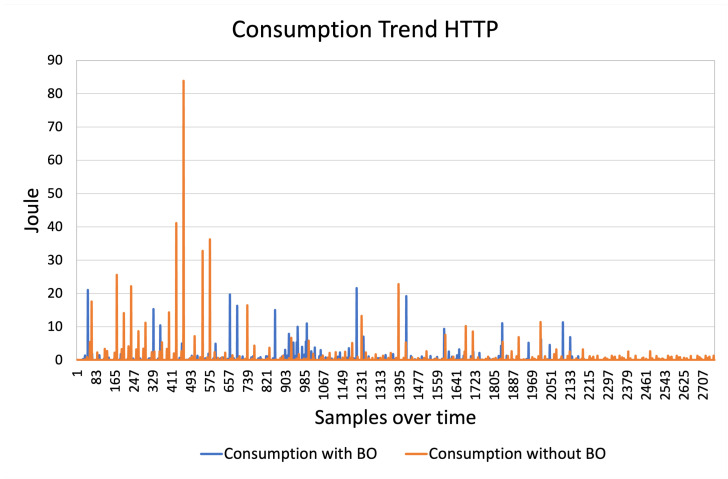
HTTP with Batch. Start Time: Monday 3 April 22:59:26, End Time: Tuesday 4 April 00:01:03, Total Consumption: 392.856065 J. HTTP without Batch. Start Time: Tuesday 4 April 13:01:24, End Time: Tuesday 4 April 14:01:57, Total Consumption: 643.59043 J.

**Figure 8 sensors-24-07246-f008:**
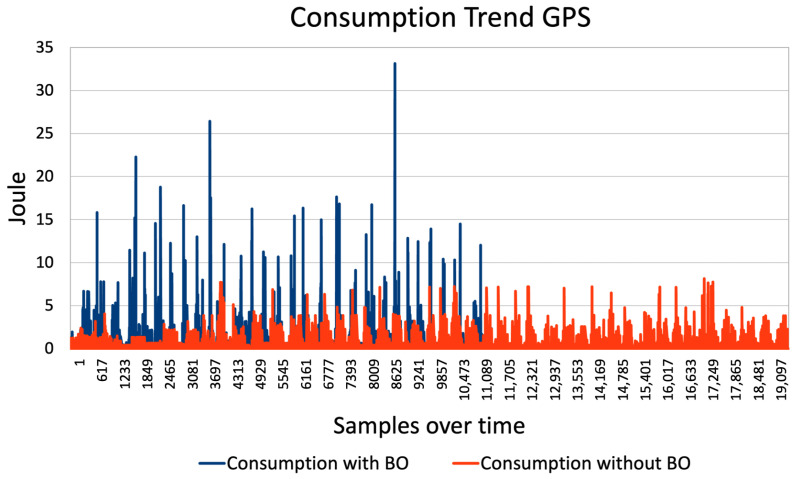
GPS with Batch. Start Time: Tuesday 4 April 21:10:57, End Time: Tuesday 4 April 22:09:32, Total Consumption: 2357.88609 J. GPS without Batch. Start Time: Friday 7 April 16:07:33, End Time: Friday 7 April 17:02:17, Total Consumption: 2266.757144 J.

**Figure 9 sensors-24-07246-f009:**
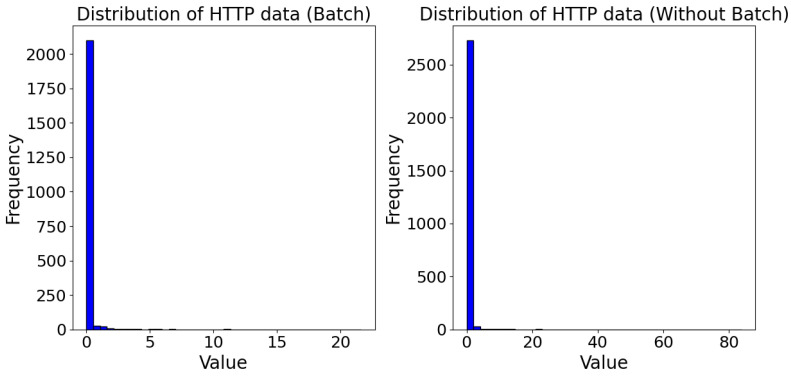
Distribution of data for HTTP with batch on the left and without batch on the right.

**Figure 10 sensors-24-07246-f010:**
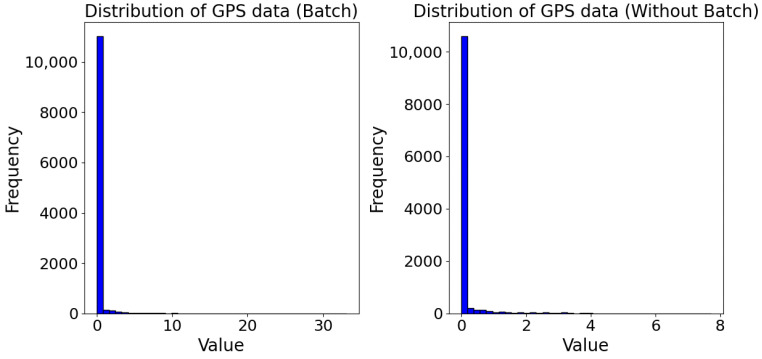
Distribution of data for GPS with batch on the left and without batch on the right.

**Table 1 sensors-24-07246-t001:** Energy consumption output given by PowDroid tool [31].

Metric	Description	Unit
Start time	when the event start	Timestamp
End time	when the event ends	Timestamp
Duration	duration of the event	Millisecond
Voltage	electric voltage emitted by the battery	Millivolt
Remaining charge	electric charge remaining in the battery	Milliampere-hour
Intensity	electric intensity calculated from remaining charge	Milliampere
Power	amount of energy during a given rime, usually 1 s	Watt
Consumed charge	amount of charge passing through the cross-section of smartphone	Milliampere-hour
Energy	total energy consumed when the application was running	Joule
Top app	name of the package running in the foreground	String
Screen	indicates if the screen is active or not	Boolean
GPS	indicates if GPS is on or not	Boolean
Mobile radio	indicates if mobile GSM is active for scanning or transmitting data	Boolean
Wi-Fi on	indicates the status of Wi-Fi (On/Off)	Boolean
Wi-Fi radio	indicates if the Wi-Fi is transferring data	Boolean

**Table 2 sensors-24-07246-t002:** 4G status signals during testing phase.

Parameter	Value
Band	2150.0 MHz
Technology	4G LTE
Signal Strength	−101 dBm
Signal Quality	15.3 dB
Latency	23 ms
Download	33.5 Mbps
Upload	34.8 Mbps

**Table 3 sensors-24-07246-t003:** GPS status signals during testing phase.

Parameter	Value
Accuracy	20 m
Number of satellites	8
Average SNR	24.7
Maximum SNR	35
Minimum SNR	18

**Table 4 sensors-24-07246-t004:** Results from Mann-Whitney U-Test for HTTP and GPS Tests. The *p*-value indicates statistical significance at the 0.01 level.

Test	Mann-Whitney U Statistic	*p*-Value
HTTP Test	68,066,280.0	0.00811
GPS Test	2,114,565.5	$4.58185\times 10^{-12}$

## Data Availability

Replication package, including source code and data, is available on GitHub [11].

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
