# Peer review of "Measuring the Effectiveness of the ‘Batch Operations’ Energy Design Pattern to Mitigate the Carbon Footprint of Communication Peripherals on Mobile Devices"

_sensors, 2024, doi:10.3390/s24227246_

Round 1
Reviewer 1 Report
Comments and Suggestions for Authors
This paper presents a study of advantages of using an energy design pattern named ’Batch Operations’ (BO) to optimize 5 energy consumption on mobile devices. Overall, the manuscript is written well and can be recommended for publication in Sensors after addressing following minor issues.
1. Why is consumption more for GPS testing vs HTTP (Fig 7 and 8) when using BO and without BO?
2. Results/discussion needs to be a separate section and conclusions should be a separate section.
3. Section 5.1 needs to be part of discussion and not conclusion.
Reviewer 2 Report
Comments and Suggestions for Authors
The article aims to experimentally verify the effectiveness of the "Batch Operations" (BO) energy design pattern in reducing the carbon footprint of communication peripherals on mobile devices.
Originality and Importance of the Study: The article proposes an interesting research topic, namely reducing the energy consumption and carbon emissions of mobile devices through software design patterns. This is a very timely and important field of research, as the energy efficiency and environmental impact of these devices are increasingly gaining attention with the proliferation of Internet of Things (IoT) devices.
Methodology: The authors employed rigorous experimental methods to assess the effectiveness of the BO pattern, including the development of Android applications, the use of energy monitoring tools, the design of mathematical models to quantify carbon emissions, and the conduct of statistical analysis. This methodology is scientific and provides strong support for the research findings.
Experimental Design: The experimental design is thorough, including tests of 4G and GPS communication interfaces, as well as comparisons with and without the BO pattern. The experiments were repeated sufficiently to ensure the reliability of the results. Additionally, the authors considered the impact of different signal strengths and the number of satellites on GPS testing, which increases the complexity of the experiment and its applicability to the real world.
Results: The study results indicate that the BO pattern can save up to 40% of energy when sending HTTP requests, thereby significantly reducing CO2 emissions. However, for the GPS interface, the BO pattern did not bring the expected advantages. These findings are very important for understanding the performance of the BO pattern in different application scenarios.
Statistical Analysis: The authors used a non-parametric hypothesis test (Mann-Whitney U-Test) to determine the statistical significance of the experimental results, which is an appropriate choice since the data distribution does not conform to the Gaussian distribution. The statistical analysis provides additional credibility to the research findings.
Discussion and Conclusion: The discussion part of the article deeply explores the potential applications and limitations of the BO pattern and proposes directions for future research. The authors also considered the internal and external validity of the study and proposed possible threats.
Overall, this article makes a valuable contribution to the study of energy efficiency and sustainability in the field of mobile devices and IoT, providing convincing evidence to support the effectiveness of the BO pattern in reducing energy consumption and carbon footprint.
Thus the manuscript may be accepted in present form.
Author Response
Thank you!
Reviewer 3 Report
Comments and Suggestions for Authors
This paper addresses an important issue in the development of smart cities: the environmental impact of the Internet of Things (IoT) and the significant energy consumption and emissions associated with mobile devices in such environments. The authors propose using a “Batch Operations” (BO) energy design pattern to optimize energy consumption, specifically for mobile devices using communication peripherals. The paper’s key contribution is an experimental validation showing the BO technique’s potential to save energy and reduce carbon emissions by up to 40% when used with HTTP requests on Android devices. However, it also concludes that the BO approach has limited applicability for GPS interfaces, adding valuable insight into the selective benefits of batch processing.
1 While the study is well-focused, the discussion lacks a broader analysis of how BO might impact other communication peripherals, such as Wi-Fi or Bluetooth, which are also prevalent in IoT devices. The limitations of the BO approach for GPS should be explored in more depth, particularly regarding potential alternative methods to optimize GPS energy consumption.
2 More details on the experimental setup would be beneficial. Specifically, the authors should describe how energy consumption was measured and what baseline conditions were used for comparison.
3 Providing information about the duration and frequency of BO applications in the experiments could enhance replicability and allow readers to assess the broader applicability of the results.
4 The discussion on carbon emission reduction could be expanded. It would be useful to translate the energy savings observed in the experiment into more tangible environmental metrics, such as estimating the annual COâ‚‚ reduction per device or projecting potential savings across large-scale implementations.
Comments on the Quality of English LanguageThe paper has a few grammatical issues and minor typographical errors. The paper would benefit from a final proofreading pass to improve readability and clarity.
